# The development of the "Laab Nuer Model" for food safety management in handling traditional Lanna cuisine in Thailand

Vivat Keawdounglek[1]*, Supat Chaiyakul[2], Anuwat Aunkham[1]

1 Program of Environmental Health, School of Health Science, Mae Fah Luang University, Chiang Rai, Thailand, 2 Department of Nutrition, Faculty of Public Health, Mahidol University, Bangkok, Thailand

* vivat.kea@mfu.ac.th

## Abstract

Food contaminants in traditional Lanna cuisine persist as a public health issue since there is no readily apparent method to prevent this contamination. Hence, this study was conducted to develop a practical strategy for reducing food hazards in traditional Lanna cuisine, according to the farm-to-fork framework across upstream, midstream, and downstream operations. In the upstream operation, 80 sample analyses in a fresh market using the chemical test-kits were conducted, and 31 samples of produce were collected to analyze them for Paraquat contamination. In the midstream operation, a biological contamination analysis was conducted on 25 of the food handlers. In the downstream operation, which refers to the consumers, this study included a questionnaire to measure the level of agreement for the different factors by using the One Way ANOVA at the 0.05 level of significance. According to the "Laab Nuer Model" for traditional Lanna cuisine, one needs to consider the following at every stage:1) *Uncontaminated market* produce, including monitoring the raw material in the market, providing the activities linked to appropriate agricultural practices, and supporting the higher price if farmers adhere to good agricultural practices; 2) *Good Restaurants* including the monitoring of their raw materials, participating in food safety practices, and providing food safety training; 3) *Paying Attention to the Consumer* including training in food safety in the younger generation and promoting food safety. To significantly decrease the impact of foodborne illness in this and other locations, this model should be implemented widely, for example, in supermarkets and flea markets.

## Introduction

Humans need food to survive. This is because food includes nutrients and minerals that are essential for the body's growth and repair, including protein, fat, and several kinds of mineral [1]. Furthermore, the nutrients are those that the body needs to be

**Data availability statement:** Data cannot be shared publicly because data contain potentially identifying or sensitive patient information, especially the name of Traditional Lanna Restaurant, and the result of the chemical and biological contamination for each restaurant. However, you can contact the human ethical committee of Chiang Rai Provical Health Officer for more information via this e-mail ssjcrhr@gmail.com.

**Funding:** This study was financially support-ed by the Office of the Permanent Secretary, Ministry of Higher Education, Science, Research, and Innovation, Grant No. RGNS 63-190, and the Research Publication Award from Mae Fah Luang University. The funders had no role in study design, data collection and analysis, decision to publish, or preparation of the manuscript.

**Competing interests:** The authors have declared that no competing interests exist.

nourished and are found in food. The body needs them to maintain overall health and physiological processes, such as energy production, muscle growth, body temperature regulation, immune system stimulation, and other vital bodily activities [2]. Even though we need food to survive, there are some cases where eating certain foods can cause symptoms to worsen annually. According to WHO data [3], each year 420,000 people die, accounting for the loss of 33 million years of healthy life, and an estimated 600 million people worldwide become hospitalized after consuming food that has been contaminated. The estimated statistics from the Centers for Disease Control and Prevention [4] state that each year, 3,000 people die from foodborne illnesses and 1 in 6 Americans contract diseases from eating or drinking contaminated food. More than $15.6 billion is projected to be lost to foodborne illnesses each year in the US economy. Around a quarter of the world's population resides in the Southeast Asia Region of the world, according to the World Health Organization [5]. As a result, reinforcing a food control system is essential for international trade and health as well as for the countries in this region. With 150 million illnesses and 175,000 deaths from food-borne illness in 2010, the Southeast Asia region has the second-highest burden of food-borne disease among all the WHO regions. Furthermore, foodborne illnesses can originate from traditional foods in Southeast Asia, which represent a category of novel foods derived from the unique cultures or ingredients of each region, encompassing plant, microbial, fungal, algal, and animal sources [6–8]. Examples of the public health problems from traditional food in Southeast Asia included in the study of Hue Vu Thi et al [9] indicate that Vietnam experienced roughly 4,000 occurrences of food poisoning due to native components such as toads, mushrooms, and insects. Moreover, inadequate handling and storage at appropriate temperatures may render traditional cuisine in Lao PDR a source of infections caused by food [10] In addition, approximately 50% of Indonesia's street food has been contaminated with chemical and biological contaminants, harming consumer health [11]. The research conducted by Khine Hsu Wai & Thunwadee Tachapattaworakul Suksaroj [12] reveal that a significant amount (70% of salads) in Myanmar is contaminated with coliform bacteria. In summary, foodborne diseases, particularly those associated with traditional cuisine in Southeast Asia, may present a significant challenge to food sanitation and safety along with public health issues.

Thailand is a popular destination for tourists who want to explore its spectacular natural resources, discover its variety of cultures, and taste its delicious food. As can be observed from the UN tourist [13] data, Thailand's GDP growth from the tourist sector from 2008 to 2021 averaged 5.5%, positioning it third in Southeast Asia for the tourism sector. Regarding the suitability of the landscape and the environment, this location is appropriate for a wide variety of agricultural crops that may provide a significant variety of food to the people of the country [14]. There are many different types of mouthwatering dishes that can be found in northern Thailand, which is a testimony to the Lanna gastronomic tradition. In addition, Tasteatlas [15] ranks "Khao Soi," a type of Lanna cuisine, as the sixth-best cuisine in the world. According to the findings of Onanong Thongmee et al. [16], traditional Lanna cuisine consists of fish, wild vegetables, and animal meat from natural sources, such as frogs and other wild

animals, which provide for a variety of tastes. Nowadays, people choose to consume fish, chicken, beef, and pork in that order of preference. While sweetness originates from specific cooking items, sugar is not a common ingredient in Traditional Lanna cuisine. Additionally, popular gastronomy from the Lanna culture is served in Chiang Rai, a border area with the People's Republic of Myanmar and Lao PDR and the origination of the Lanna Kingdom from the twelfth century [16], for example, at round dining tables known as Khantoke, Khanom Jeen Nam Ngiao or noodle curry-soup, Khao Soi or Northern noodle curry, and Nam Prik Noom or chili-drips are served.

However, the primary factor influencing the epidemic of foodborne illness in traditional Lanna cuisine is due to the lack of food safety management, including inappropriate handling of supplies or raw materials, poor personal hygiene, and a lack of food surveillance [17–18]. As evidenced by the annual epidemiological surveillance report for Thailand for 2021, Chiang Rai is the sixth highest province in Thailand where there is a risk of food poisoning, with approximately 208 cases per population of 100,000 [19]. According to a study conducted by Keawdounglek [20], 50% of local food restaurants in northern Thailand detected *Escherichia coli* (*E. coli*) contamination on chefs' hands, utensils, and certain drinking water sources, despite the implementation of food handler registration in 2018. The absence of risk analysis regarding food hazards in traditional Lanna cuisine may be obvious and there is also a lack of studies analyzing the risks of food hazards within the farm-to-fork framework of traditional Lanna cuisine [21,22]. This includes upstream operations related to fresh markets supplying ingredients for this cuisine, which could present food hazards, particularly concerning chemical contamination in raw materials [23]. The subsequent part is the midstream operation designated for food handlers in this cuisine, they may lack knowledge of good personal hygiene and sanitation practice [24]. The final stage in the downstream operation concerning customer involvement in traditional Lanna cuisine should be educational training programs that enhance their knowledge of food safety, and the receiving of food safety information through food safety certification labels [22,25]. There may be environmental health issues for tourists and others who consume traditional Lanna cuisine which may be contaminated with pathogens or other harmful substances [21] caused by insufficient food safety management of traditional Lanna cuisine.

To date, no prior studies have applied this food safety concept within the context of local Northern cuisine. The selection of "Laab Nuer," a traditional dish (see Fig 1) with deep cultural roots dating back to the Lanna Kingdom, which originated from regional vegetables in most instances from the northern area, such as ground pork, and specific seasoning, for example, Lanna chili paste ("Laab Nuer") which the peoples in northern Thailand believe provides prosperity and

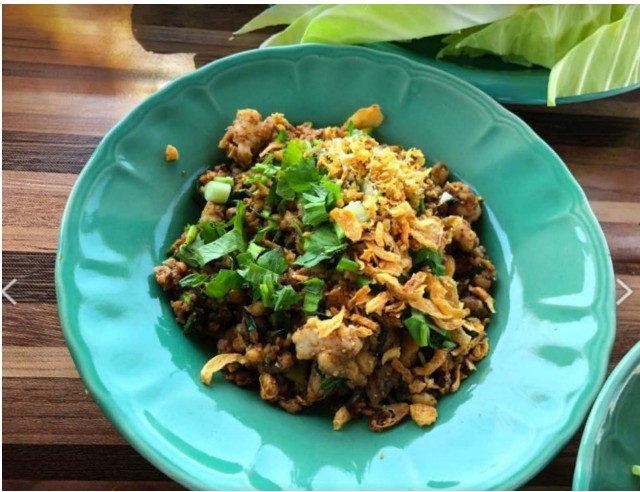

**Fig 1. Laab Nuer[a]. [a] Photographed by Vivat Keawdounglek.**

accomplishment in many undertakings [16,26]. In addition, the preparation of Laab Nuer involves purchases of supplies from markets or specialized retailers, including meat, fresh vegetables, and spices, which are essential for the quality of the more effective resources from *upstream* [27]. After holding quality ingredients, the subsequent phase is termed *the midstream*, in which whoever is preparing the recipe must adhere to appropriate personal hygiene practices to prevent food-borne diseases associated with consuming Laab Nuer [28]. Eventually, customers, including the local population and both Thai and foreign tourists, who represent the *downstream*, will have access to Laab Nuer that is clean and fit for consumption [29]. Consequently, the development of the "Laab Nuer Model" in this study constitutes an adaptation of the farm-to-fork food safety management framework, which emphasizes comprehensive food safety oversight across the entire supply chain, from production (upstream), through processing and distribution (midstream), to the consumers (downstream) [21,22,24,25].This model was specifically designed to align with the cultural and dietary context of Northern Thai traditional foods. Furthermore, the development of this model corresponds with the farm-to-fork strategy established by the European Commission [30], which argues that this principle ensures food security, nutrition, and public health, ensuring general access to adequate, safe, nutritious, and sustainable food. In addition, this model aligns with the principles of Hazard Analysis and Critical Control Point (HACCP), which has been considerably improved for food processing and manufacturing; it is applicable to all parts of the food supply chain, incorporating procurement, storage, cooking, and serving [21]. It is not only focal point for implementing food safety practices, but also a cultural conduit for promoting awareness and preservation of Lanna culinary heritage within public health discourse [31]. Moreover, it may also decrease hospitalization costs for foodborne illnesses in Thailand by up to $1 million per year [32]. The aim of this research, it is to develop a practical strategy for reducing food hazards in traditional Lanna cuisine, according to the farm-to-fork framework across upstream, midstream, and downstream operations.

## Materials and methods

A cross-sectional design [33], as shown in Fig 2, was carried out for this study across a variety of target groups, specifically upstream, midstream, and downstream operations, encompassing the farm-to-fork framework [21–25,30,34]. The first focus of the study group for upstream operations was a central fresh market that distributes various raw materials for the preparation of traditional Lanna cuisine. The second involved the midstream operation, which included the holders and food handlers in a traditional Lanna restaurant for a target group to identify biological hazards and contaminants. The final

**Farm-to-Fork framework for model development**

| Upstream Operation | Midstream Operation | Downstream Operation |
|---|---|---|
| • Target group was a *fresh market* <br> • Research methods were the *chemical test-kits analysis* for 80 samples and *paraquat analysis* for 31 samples | • Target group was *25 of traditional Lanna restaurants* <br> • Research methods were the *observation* and the *assessment of biological contamination* | • Target group was *305 of consumers* <br> • Research method was the *analysis of the agreement level for this model using the questionnaire* |

**Fig 2. Summary of the research methodologies.**

group comprised the downstream operation, consisting of customers who had previously eaten traditional Lanna cuisine in the area, categorized as the downstream operation. In addition, this study obtained approval from the Ethics Review Committee for Research Involving Human Subjects at the Chiang Rai Provincial Health Office (experimental protocol no. CRPPHO 131/2564), and all the participants provided written informed consent before participating in the study. Moreover, the researcher explained all the details to the volunteers and obtained their signatures on a consent form to enable work on the study to proceed. The methodologies for upstream, midstream, and downstream operations have been defined as follows:

## Method for the upstream operation

As the primary source of raw materials for traditional Lanna cuisine in Chaing Rai, the upstream of this study was the central fresh market. Over hundreds of sellers are present in this market. To obtain corroborating data on food sanitation and safety in traditional Lanna cuisine, an in-depth interview with the manager and staff were conducted at this source [35]. In routine practice, preliminary screening for paraquat residues at this fresh market is performed using the GT 30 test kit [36]. This screening tool enables the detection of chemical contamination in indigenous vegetables and determines whether the contamination levels fall within acceptable safety thresholds. Based on the market's recent testing data, paraquat residues were detected in 31 samples of which 9 were leaf-eating plants (i.e., lettuce, cabbage), 16 edible plants (i.e., galangal, tomato, black pepper), and 6 stem-eating plants (i.e., long beans, coriander, and some herb); however, all of these were found to be within the permissible safety limits. Therefore, the high-performance liquid chromatography (HPLC) test from the Water, Alliance e2695 [37] was conducted for Paraquat contamination to confirm the concentration in the samples. The specifications of the Paraquat analysis procedure were as follows: 1) the major detector for the analysis was a UV DAD, and the column was a SunFire ™ C-18 column made in Ireland [38]; 2) the sample injection was performed with 10 microliters; 3) A special reagent for the mobile phase was adapted by Sithiporn Associate Co.Ltd. [39] and Guiyan Yuan et al. [40]; 4) The sample extraction followed the method of Chuang et al. [41]; and 5) the concentrations of Paraquat standards were 0.00125, 0.00375, 0.005, 0.0075, and 0.01 part per million (ppm). The equation to calculate the concentration of Paraquat in this study is as follows:

$$y = 4.039x10^4 X + 6.118x10^4 \qquad (1)$$

Where $y$ is the peak area for Paraquat analysis.

$X$ is the concentration of Paraquat (mg/kg).

$R^2$ or the coefficient of determination in this equation is 0.99795.

Additionally, this methodology applied test kits to assess the chemical risks, described as GPO ™ made in Thailand, in 80 randomized samples from this market, in accordance with the Thailand Healthy Market standards [42]. These results included: 1) GPO ™ borax detected in samples of ground pork, meatballs, and poultry; 2) GPO ™ formalin identified in samples of animal entrails; 3) GPO ™ salicylic acid present in curry paste and garlic; and 4) GPO ™ sodium hydrosulfite found in samples of bean sprouts and pickled bamboo shoots [43]. Each food sample was analyzed in triplicate (n = 3 replicates) to ensure analytical accuracy.

## Method for the midstream operation

The stall holders and food handlers in a traditional Lanna restaurant, served as the study's midstream operation. This study focused on the Lanna restaurants in Chiang Rai, Thailand which is where Lanna food and culture originated [44]. A purposive sampling of traditional Lanna restaurants was conducted [45](Eshenaur Spolarich, 2023) for this study design according to two main criteria of purposive sampling. The first criteria was the zoning as Chiang Rai has 7 administrative zones according to the Thai government gazette [46]. The last criteria is the size of a restaurant according to the study of

Phuanpoh [47] which stated that there are three groups of restaurant sizes including food stalls, small restaurants (less than twenty seats for dining), and large restaurants (a maximum of 20 seats for dining). As a result, traditional Lanna restaurants for this sampling were 21 restaurants. However, there were 25 locations of traditional Lanna restaurants, as can be seen in Fig 3, which participated in this study including 9 food stalls, 8 small restaurants, and 8 large restaurants. The inclusion criteria were used to choose the participants; 1) When asked for their responses, participants had to be able to answer and communicate in Thai, and 2) local restaurant owners or food handlers must have been operating for at least a year. If the restaurant owner or food handlers could not respond to the complete questionnaire (see S1 Table.), they were excluded. According to the American Society for Microbiology [48], *E. coli* contamination on the chefs' hands and food kitchenware can be transferred to the Eosin Methylene Blue (EMB) agar from HiMedia Co.Ltd, India. Furthermore, the assessment of biological hazards was carried out using the most probable number (MPN) technique on ice and drinking water samples in the traditional Lanna restaurants [49–50]. As regards the standard of *E. coli* contamination, this study followed the standards of the Ministry of Public Health [42–43] which should demonstrate that there should not be any *E. coli* contamination on ice and drinking water, food kitchenware, or the chefs' hands. In addition, an in-depth-interview of the traditional Lanna restaurants holders [51] was conducted to find out the initial factors affecting to the food safety management in the traditional Lanna cuisine. Each sample was analyzed in triplicate (n = 3 replicates).

## Method for the downstream operation

Customers who had previously consumed traditional Lanna cuisine in the area are referred to as the downstream operation in this study. As a result, the G*Power program determined 305 sample sizes at the 95% confidence level and 0.25 effect sizes [52]. Then, the questionnaire was developed to make a final version and distributed to 305 participants (as can be seen in S2 Table.) from the information received from depth-interviews from 25 stall holders of traditional Lanna restaurants and in the fresh market, and with the agreement of the experts of the Faculty of Public Health, Burapha University, Faculty of Public Health, Mahidol University, and the Faculty of Public Health, Thammasart University, Lampang Campus. Three topics were included to elicit information from the consumers, including 1) Food safety management at the upstream level such as random checks of fresh market produce, education to generate safe food, and the purchase of safe farm products at above-market pricing; 2) Food safety management at the midstream level such as sampling from restaurants for contamination assessment, organizing local food safety education, and establishing a food safety

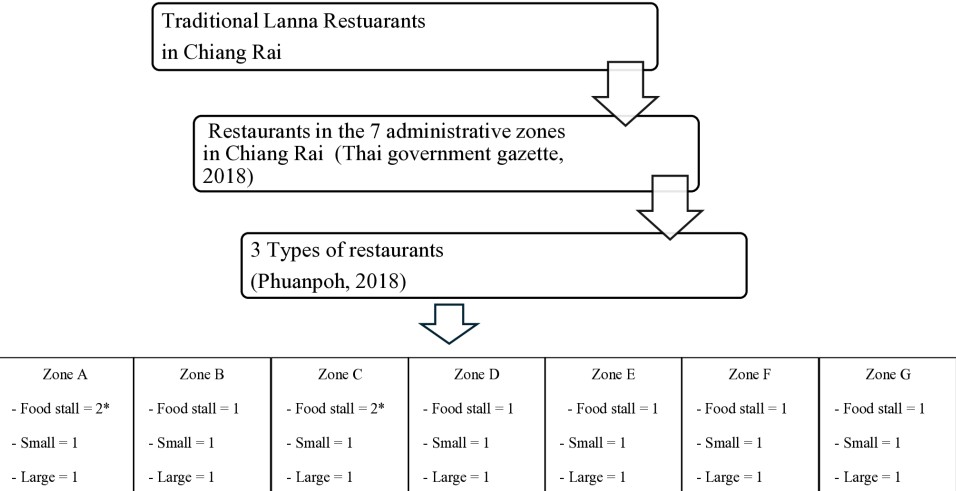

**Fig 3. Purposive Sampling of Traditional Lanna Restaurants. Note:** In Zones A and C, two of food stalls were accepted as participants in this study.

committee; and 3) Food safety management at the downstream level, such as promoting food-safe establishments, signs indicating a restaurant's safety and hygiene, promoting food safety through fairs or exhibitions, and raising food safety awareness and knowledge among students and local teenagers. The item objective congruence (IOC) test was then used to validate the customers questionnaire with the objectives of this study by 3 environmental health experts from the School of Health Science, Mae Fah Luang University. These experts assessed each consumer questionnaire based on criteria congruence, using a score of (−1) to indicate unsuitability. This study determined questionnaires with an IOC value of 0.5 or higher based on the experts' average score. The analysis had an IOC greater than 0.5, therefore, all criteria were suitable for the customer survey [33]. Moreover, the Cronbach Alpha index was then used to measure the reliability with the IBM-SPSS program version 20.0 to confirm that the results related to the objectives of this study [53]. According to the study of Edelsbrunner et al. [54], which recommended that the Cronbach Alpha index should be greater than 0.70, this analysis showed that the questionnaire had a high reliability with a Cronbach Alpha value of 0.904. The significance of factors obtained from the consumers regarding the food safety management in traditional Lanna cuisine were interpreted by the average, p-Value from One-Way ANOVA, and Eta Square ($\eta^2$) which responded to the effect size using the IBS-SPSS version 20.0 [55–56]. With regard to a normality test, all the variables had a p-Value of Kolmogorov–Smirnov test more than 0.05.

## Results

### Results for the upstream operation

From the in-depth interviews of the owner and the staff at the central fresh market, the market was created t project to support good agricultural practices to avoid harmful chemical usage when planting. Random checks are conducted on fruit and vegetables which can contain pesticides from pesticides. It was found that 82.61 percent of the samples that were inspected at this fresh market revealed no harmful residues, while some samples showed toxic residues of up to 17.39% (see in S3 Fig.). Furthermore, another project at this fresh market was concerned with promoting higher prices for suppliers who can guarantee the safety of their food. Additionally, the results of the HPLC analysis of the samples in indicated that the concentration of Paraquat in all the samples of this study at the central fresh market was lower than the CODEX standard [57], which is less than 0.005 ppm. As shown in Fig 1, most of the samples tested had Paraquat concentrations between $1.00 \times 10^{-4}$ and $5.99 \times 10^{-4}$ ppm. In addition, the concentrations of Paraquat for leaf-eating plants, edible plants, and stem-eating plants were 0.0000819, 0.0000803, and 0.0000654 ppm, respectively, as shown in Fig 4. According to the standards required by the Thailand Health Market [43], the chemical contamination test-kits found that all samples of this fresh market, including those in the upstream level, contained chemical contaminants, including borax, salicylic acid, formalin, and sodium hydrosulfide.

### Results for the midstream operation

The study of upstream operations concerning raw materials from the fresh market demonstrates that chemical contamination is acceptable under Thai standards. Nonetheless, in this study, *E. coli* was found on the hands of the chefs at about 13 sites. Similarly, samples of kitchenware, ice, and drinking water from six traditional Lanna restaurants were confirmed positive for *E. coli*. At the initial meeting, the researcher conferred with the restaurant owner and experts to determine how to resolve the problems of contamination. including proper hand washing techniques, good personal hygiene, and guidelines for handling raw materials. After a month of implementation, it was discovered that, as shown in Fig 5, the number of *E. coli* contaminations on the chefs' hands decreased from 13 sites to a total of 6 sites or 53.84%. At the same time, there was a slight shift in the number of *E. coli* contaminations of the kitchenware, decreasing from 6 to 5 sites or 16.67%. However, the amount of *E. coli* contamination remained constant on the samples of ice and drinking water.

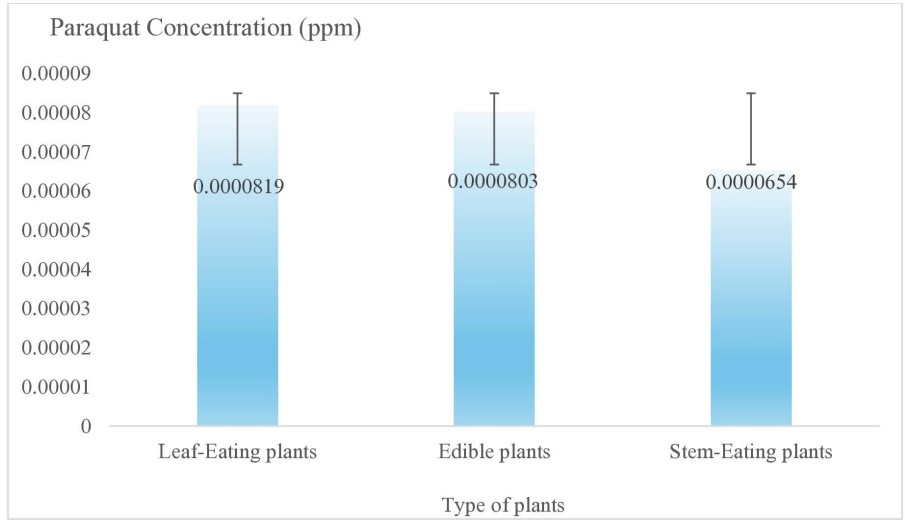

**Fig 4. A comparison of Paraquat concentrations [a] in various types of plants.** [a] The CODEX [57] requires that the Paraquat standard limit be less than 0.005 ppm.

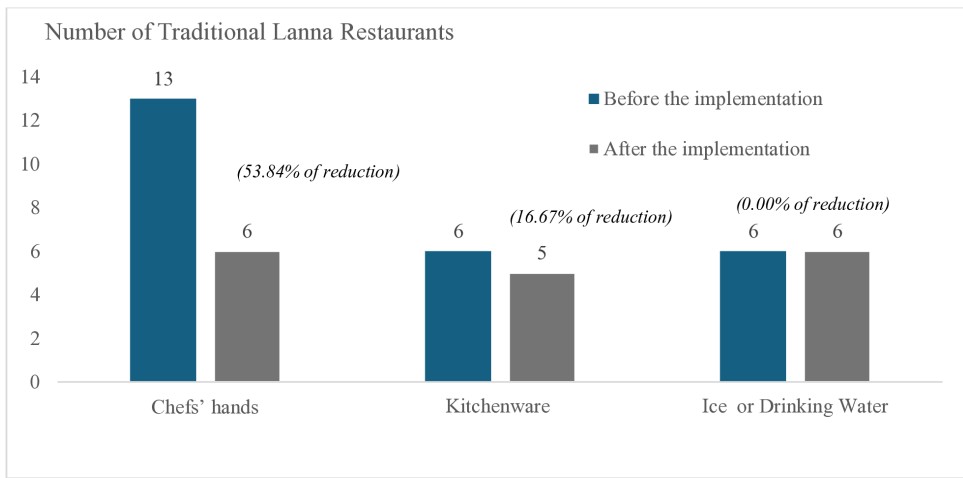

**Fig 5. A comparison of *E. coli* contamination at Traditional Lanna restaurants before and after the model's implementation.**

## Results of the downstream level

According to a One-Way ANOVA test (see in S4 Table.) between the food safety management at the upstream and the overall score for food safety management in traditional Lanna cuisine from the consumer, it was found that all of the items including random checks of fresh market produce ($\overline{X}$ = 4.65 from 5, p-Value < 0.05, $\eta^2$ = 0.305), education to generate safe food ($\overline{X}$ = 4.58 from 5, p-Value < 0.05, $\eta^2$ = 0.334), and purchasing safe farm products at above-market pricing ($\overline{X}$ = 4.28 from 5, p-Value < 0.05, $\eta^2$ = 0.334), received are a high average rate of agreement from the consumers. Moreover, the results are statistically significant at the 0.05 level for all items. In addition, the Eta Square for all variables have more than 0.14 indicating a large effect size for all variables. This suggests that upstream processes, such as random inspections of vegetables, fruits, and raw materials in fresh markets for potential contamination, farmer education on consumer-safe

produce, and purchasing produce from farmers practicing safe methods at prices above the market rate. are crucial for the food safety management of traditional Lanna cuisine.

With regard to the food safety management at the midstream level, the overall scores for food safety management in traditional Lanna cuisine for all items including sampling from restaurants for contamination assessment were: $\overline{X}$ = 4.65 from 5, p-Value < 0.05, $\eta^2$ = 0.368), organizing the local food safety education ($\overline{X}$ = 4.58 from 5, p-Value < 0.05, , $\eta^2$ = 0.410), and establishing a food safety committee ($\overline{X}$ = 4.59 from 5, p-Value < 0.05, $\eta^2$ = 0.443), all of which represent a high level of agreement from the consumers. Moreover, the results were statistically significant at the 0.05 level for all items. In addition, the Eta Square for all variables was more than 0.14 indicating a large effect size for all variables. Therefore, it was demonstrated that the midstream processes in food safety management include collecting restaurant food samples to test for chemical and microbial contamination, organizing community-based food safety education activities, and establishing a provincial food safety club or committee midstream interventions succeeded in developing a safe and comprehensive food safety management in traditional Lanna cuisine across the food chain.

The last part of the consumer survey was at the downstream level. The findings indicated that all items were related to the promotion of food safety in establishments ($\overline{X}$ = 4.52 from 5, p-Value < 0.05, $\eta^2$ = 0.355), such as signs indicating a restaurant's safety and hygiene ($\overline{X}$ = 4.62 from 5, p-Value < 0.05, $\eta^2$ = 0.369), promoting food safety through fairs or exhibitions ($\overline{X}$ = 4.50 from 5, p-Value < 0.05, $\eta^2$ = 0.284), and raising food safety awareness and knowledge among students and local teenagers ($\overline{X}$ = 4.58 from 5, p-Value < 0.05, $\eta^2$ = 0.331). All of these were significantly related to the overall score for food safety management in traditional Lanna cuisine as shown by the less than 0.05 for p-Value. Moreover, the Eta Square value ($\eta^2$) for all variables exceeded 0.14, indicating a large effect size.

## The final model for food safety management for traditional Lanna cuisine

The development of this model requires the application of the principles of Thai public administration. Typically, Thai public administration contains two components for policy implementation: 1) the central government, which includes ministries, departments, the Prime Minister's Office, the National Economic and Social Development Board, and the Budget Bureau. The latter refers to the local government, encompassing regional organizations, provincial organizations, and municipalities [58–60]. After the study of food safety management in the traditional Lanna cuisine, we can summarize the results as can be seen in S5 Table. For the upstream operation, the fresh market promotes good agricultural practice (GAP) to prevent any chemical hazard on their raw materials, and therefore their raw materials passed a Thai standard for chemical contamination. However, during in-depth interviews with the stall holders at the fresh market, they said that they lacked support for higher production costs, even if certain farmers had proof that their products were safe from chemical contamination. Therefore, the monitoring of the raw material in the market, the encouragement of farmers to participate in the activities related to appropriate agricultural practices, and the support for higher production prices for farmers who can certify the safety of their products should be a collaborative effort involving the central government, local government, and the owner of this fresh market. In midstream operations, both the stall holders and food handlers of traditional Lanna restaurants actively engage in adhering to food safety principles and good hygiene practices. However, several restaurants lack food safety assessments, and some food handlers have never received training in food safety for the restaurant sector. Consequently, the local government, restaurant and food stall holders and food handlers ought to monitor their raw materials and support the relevant sector to complete or pass food safety training and engage in food safety initiatives. The last operation is at the downstream level which focuses on the customers. The results indicated that all variables achieved a high average score of less than 0.05 of p-Value in an ANOVA analysis, and provided a more than 0.14 of Eta squared indicating a large effect size (as can be seen in the results of the downstream operation). However, several consumers indicated a lack of understanding of food safety management practices related to traditional Lanna cuisine. Consequently, both national and local governments should advocate for a commitment to food safety among consumers and provide training on food safety and sanitation for the younger generation.

Therefore, a summary of the research findings recommends practical strategies for important stakeholders, as listed below:

1. *Uncontaminated Market* which is the traditional Lanna cuisine at the upstream level. They should monitor the raw materials in the market, particularly those that may be contaminated with chemicals including pesticides, borax, salicylic acid, sodium hydrosulfide, formalin, and others. Moreover, they should encourage some farmers to participate in projects or activities linked to appropriate agricultural practices, such as controlling pests and properly using chemicals on their land. In addition, they should support the high value of production price if the agriculturists claim that their products are safe for customers. Furthermore, central government, local government, and the stall holders of the fresh market should engage in these activities.

2. *Good Restaurants* at the midstream level should monitor their raw materials and encourage the relevant sectors, including the local government administration or the local hospital to send samples for examination for biological and chemical contamination. Moreover, they should participate in food safety activities, and they need to complete or pass food safety training. Moreover, there are two sectors which should engage in this process including the local government, and the traditional Lanna restaurants.

3. *Attention should be paid to consumers*. To extensively publicize the food sanitation and safety based on traditional Lanna cuisine, food safety signs should be displayed and restaurants which have a food safety certificate should be promoted. Furthermore, the younger generation, such as adolescents and children, should receive training on food safety and sanitation so that they may educate parents or other people at the community level. In addition, the central and local government should focus on this process to increase customer participation.

   Fig 6 summarizes the concept of "Laab Nuer Model" for food management in traditional Lanna cuisine at Thailand.

## Discussion

In the first section of the "Laab Nuer Model", the monitoring of raw materials in the market is necessary for food safety management because some pesticides such as chlorpyrifos, profenofos, tebuconazole, diazinon, and fipronil are frequently found, although persistent organic pollutants (POPs) which might contaminate some of the raw products are less common, including several kinds of vegetables that are used for traditional Lanna cuisine [61]. Moreover, the criteria for determining the chemical contamination of the raw materials at markets in Thailand are linked to the established requirements for healthy market standards, including 1) Borax present in samples of pork balls, ground pork, etc.; if the sample includes tainted borax, which displays a shade of red formed on the cucumber paper. In addition, in Bangkok, a test was conducted for borax from meatball samples. It was found that the number of samples contaminated with borax substance was up to 60.47%, especially fish balls which contained the most [62]; 2) Formalin was found in samples of vegetables, fish [63] and animal guts; the detection of red in a test-kit solution indicates that the materials are contaminated with formalin. In Somdet Market, Kalasin Province, Thailand, they randomly test seafood and frozen meat for formalin contamination. It has been found that all fresh seafood samples were contaminated with formalin [64]; 3) salicylic acid, which, in cases of contamination appear to be black or violet in color in tests of pickled fruit, curry paste, sour pork, and aquatic products [65] etc. Additionally, there are sporadic inspections for salicylic acid contamination in fruits, vegetables, and pickled fruit in Tambon Mamuang, Songton, Nakhon Si Thammarat, Thailand. Salicylic acid is a contaminant which is detectablw when random samples are checked for contamination in vegetables and pickled fruit, which account for percentages of 51.14 and 60.00 [66]; 4) sodium hydrosulfite tests, such as pickled bean sprouts, bamboo shoots and sugar (Sifan Wang, 2024); these show a black or gray color in appearance if the sample is contaminated. In 2023, using a contamination test kit, samples were gathered for the investigation of sodium hydrosulfite contamination in some fresh markets in Bangkok and the surrounding areas. A total of 100 food samples were tested, 20 of each of the following five food

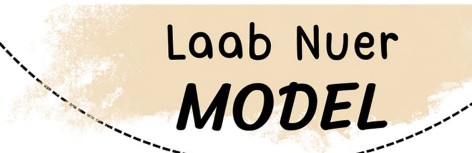
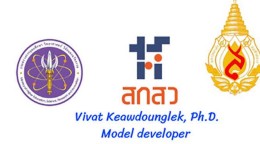
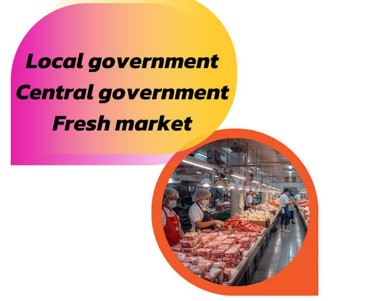

**Fig 6. "Laab Nuer Model" for food safety management in handling traditional Lanna cuisine**[b]. [b] Image generated using Canva AI (https://www.canva.com), licensed under Canva's Content License Agreement (https://www.canva.com/policies/content-license-agreement/).

categories: bread, pickled bamboo shoots, fish balls, noodles, and rice. The result concluded that 20% of the total noodle products, or 4 samples altogether—2 samples of rice vermicelli and 2 samples of noodles—were in the 10–100 ppm range [67]; additionally, 5) pesticides present in a variety of fruits and vegetables, including Chinese kale, cabbage, long beans, and other vegetables; when the MJPK test-kits were applied to samples that were extremely contaminated with pesticides, the results showed a pinkish-orange color [42. Moreover, a project for good agricultural practice or GAP provides a guarantee of safe products at farms which encourages food safety at the upstream level. At every point of the food chain, from primary production to secondary and tertiary processing, storage and distribution, and packaging, there may be risks of food contamination. Therefore, it is crucial to address food safety management beginning at the agricultural level. Ensuring a safe food supply greatly depends on implementing best practices during the on-farm and post-production phases. According to the Food and Agriculture Organization [68], good agricultural practices (GAP) are a collection of principles to apply to on-farm production and postproduction processes, resulting in safe and healthy food and non-food agricultural products, while considering economic, social, and environmental sustainability.

The next section deals with *"Good Restaurants"* which is for traditional Lanna restaurants. In fact, food safety training in restaurants performs a critical role in reducing contamination and biological hazards. As can be seen in the number of *Escherichia coli* contaminations on the hands of chefs, the number of restaurants contaminated with *E. coli* decreased after the implementation of the model from 13 to 53%. Wang et al. [69] states that they might have billions of pathogens on their hands. According to a study by Honggang Lai et al. [70], proper hand washing and addressing fundamental sanitation procedures can help reduce biological contamination, including foodborne illnesses and *Campylobacter*. To protect customers from foodborne infections, a monitoring program of restaurants' raw materials is necessary, including the handling of ice and drinking water samples, although the World Health Organization [71] determined that they could not be contaminated with *E. coli*. Furthermore, participating in food safety training may improve the restaurant owners' attitudes

and practice of personal hygiene because they can increase the food workers' knowledge of food hazards and teach them how to avoid problems traditional Lanna restaurants. Therefore, food workers should modify their cooking and food preparation methods by maintaining proper personal hygiene [72].

The final section of this model, *"Attention should be paid to the consumers"*, because the customers are essential in advancing food safety management in a traditional Lanna restaurant at the downstream level. The study conducted by Hong Phuc Luu et al. [73] suggests that many factors connected to consumer behavior should be addressed for the successful implementation of food safety management. These factors include exposure to food safety guidelines, choice of lifestyle, and social influence. It is possible for the younger generation to become aware that outbreaks of foodborne illness can be caused by physical, chemical, or biological food contamination (Perilli et al., 2023). Therefore, it is important to train the younger generation in food safety following this model so that after acquiring this knowledge, they might then pass it on to their parents, which could alter family perspectives on food sanitation and safety [29]. Moreover, food safety management at the downstream level may be impacted by the promotion of food safety exhibitions because time spent at these events by visitors can provide information about the significance of food sanitation and safety and offer a preliminary understanding of the attractions of traditional Lanna cuisine [74]. Furthermore, restaurant ratings according to food safety guidelines is an effective way to extensively advertise traditional Lanna cuisine and restaurants because it positively impacts on consumer perceptions of food safety management [75].

This model aligns with global best practices that emphasize risk-based food safety management, intersectoral collaboration, and community empowerment, as shown in the WHO's Five Keys to Safer Food [34], the CODEX Alimentarius Commission guidelines [76] and the European Commission's Farm-to-Fork strategy [30]. Moreover, routine traditional Lanna restaurant sampling and contamination testing follows the Hazard Analysis and Critical Control Points (HACCP) system used in food regulatory systems worldwide [17]. From this information, food safety management across the supply chain requires starting with the production of safe raw materials, compliance to adequate hygiene practices by food handlers, and the recognition of consumers as direct stakeholders who are impacted by the consumption of contaminated and unsafe food, relating to the principle of HACCP that focuses on the critical control point in the traditional Lanna cuisine. Despite this model not promoting the documentation and the regulatory oversight as the HACCP's principle, the "Laab Nuer" model can be relied upon for the building of knowledge and awareness for food safety in traditional Lanna cuisine, which is the same principle of five keys and food safety from the World Health Organization. FAO/WHO food safety capacity-building programs emphasize stakeholder engagement and culturally relevant health communication, which is reflected in localized educational campaigns to promote safe food practices [77]. In addition, the One Health framework emphasizes multisectoral coordination in managing food-related health risks, and the food safety committee strengthens decentralized governance and allows responsive, context-specific strategies [78]. Finally, the "Laab Nuer" model can improve local capacity for food safety in traditional food systems and it is compatible with internationally endorsed models, contributing to the public health goal of reducing foodborne illness and promoting food safety management.

Furthermore, this study some recommendations for additional research. Firstly, this Laab Nuer Model can manage food safety in traditional Lanna cuisine. Nonetheless, some food categories, including street foods and products from flea markets, are excluded from this model's application. As a result, a variety of food types, that includes food establishments, street vendors, and products from flea markets, should be mandated by this model to avoid food contamination and foodborne illnesses, which represent risks to people as well as visitors, potentially affecting Thailand's tourism and, by extension, its future economic growth [79]. Secondly, this study does not provide a policy and stakeholder analysis, particularly concerning central government. Therefore, such an analysis of food safety management policies and stakeholders should be incorporated in a further study [80–83] so that the results can inform central policymakers, thereby enhancing the sustainability of food safety management by broadly promoting a positive view of public health in Thailand.

## Conclusion

Traditional Lanna cuisine is the hallmark of northern Thailand's culinary culture, attracting huge numbers of tourists seeking to sample the food's delicious ingredients, which include various herbs and Lanna seasoning. However, Chiang Rai is listed as the sixth highest location regarding the risks of customers suffering from acute diarrhea and food poisoning. Therefore, the "Laab Nuer Model" needs to be researched and developed to promote the economic growth derived from traditional Lanna cuisine and to prevent the main contributory factors of the risks of food contamination and foodborne diseases from traditional Lanna cuisine.

This study concludes that the central fresh market at the upstream level has been monitoring any contamination that might present a health risk to consumers in their raw materials. According to the Paraquat analysis, every sample had less Paraquat than is required by CODEX and Thailand's regulations. At the midstream level, the researcher is collaborating with the local public health officer and local government administration to develop a training program for food workers on proper personal hygiene. There will be a 50% reduction in *E. coli* contamination on their hands following this training and a month of implementation. In addition, all the factors that relate to the consumers at the downstream level will have a significant effect on food safety management for the traditional Lanna cuisine at the 0.05 significance level.

As part of the "Laab Nuer Model," the serving of traditional Lanna cuisine should be addressed at all levels, from the upstream to the downstream levels as follows: 1) *Uncontaminated foods in the market*; 2) The *Good restaurants*; and 3) *Attention should be paid to the consumers*. Additionally, this model should also be applied to other food sources, such as supermarkets and flea market products, to significantly mitigate the effect of foodborne diseases in the north-east and in other areas of Thailand.

## Supporting information

**S1 Table.  Questionnaire for the owner or the food handler in the traditional Lanna restaurant.**
(PDF)

**S2 Table.  Questionnaire for the downstream.**
(PDF)

**S3 Fig.  A comparison of Paraquat contamination[a] numbers and percentages in different concentrations (unit:ppm).**
(TIF)

**S4 Table.  The statistical testing from the consumer from the IBM SPSS version 20.0.**
(PDF)

**S5 Table.  Summary of the findings, practical strategies, and responsible persons.**
(PDF)

## Acknowledgments

The author would like to thank the participants in this study who volunteered to give their time for the data collection. Additionally, the author of this paper is grateful to Mae Fah Luang University for its assistance and support for a scholarship which enabled the publication of this article.

## Author contributions

**Conceptualization:** Vivat Keawdounglek.

**Funding acquisition:** Vivat Keawdounglek.

**Investigation:** Vivat Keawdounglek.

**Methodology:** Vivat Keawdounglek.

**Project administration:** Vivat Keawdounglek.

**Resources:** Vivat Keawdounglek.

**Supervision:** Vivat Keawdounglek, Supat Chaiyakul.

**Validation:** Vivat Keawdounglek, Supat Chaiyakul, Anuwat Aunkham.

**Writing – original draft:** Vivat Keawdounglek, Anuwat Aunkham.

**Writing – review & editing:** Vivat Keawdounglek, Supat Chaiyakul, Anuwat Aunkham.

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
