## [Decision Letter · Decision Letter 0]

4 Dec 2024

Dear Dr. KEAWDOUNGLEK,

Thank you for submitting your manuscript to PLOS ONE. After careful consideration, we feel that it has merit but does not fully meet PLOS ONE’s publication criteria as it currently stands. Therefore, we invite you to submit a revised version of the manuscript that addresses the points raised during the review process.

**ACADEMIC EDITOR: Please insert comments here and delete this placeholder text when finished.**

**One or more of the reviewers has recommended that you cite specific previously published works. Members of the editorial team have determined that the works referenced are not directly related to the submitted manuscript. As such, please note that it is not necessary or expected to cite the works requested by the reviewer. **

publication criteria  and not, for example, on novelty or perceived impact.

We look forward to receiving your revised manuscript.

Kind regards,

Mahmood Ahmed

Academic Editor

PLOS ONE

Journal Requirements:

4. In the online submission form, you indicated that all the data is available on the request

5. Please ensure that you refer to Figure 4 in your text as, if accepted, production will need this reference to link the reader to the figure.

Additional Editor Comments:

Major revision

Reviewers' comments:

Reviewer's Responses to Questions

**Comments to the Author**

1. Is the manuscript technically sound, and do the data support the conclusions?

Reviewer #1: Yes

Reviewer #2: No

2. Has the statistical analysis been performed appropriately and rigorously?

Reviewer #1: Yes

Reviewer #2: I Don't Know

3. Have the authors made all data underlying the findings in their manuscript fully available?

Reviewer #1: Yes

Reviewer #2: Yes

4. Is the manuscript presented in an intelligible fashion and written in standard English?

Reviewer #1: Yes

Reviewer #2: No

Reviewer #1: Dear Authors

The present manuscript is an interesting article with a suitable methodology, but some details should be discussed more

What is the basis of naming the model?

Does theoretical literature need more explanation?

Information about the studied groups should be mentioned in the results section.

Research limitations and recommendations for future research should be mentioned.

Reviewer #2: General:

- The language in this article needs reviewing, as it's often unclear and not written in an academic tone.

- There is a great degree of presumed knowledge, and terms need to be better explained.

- The purpose of the research and its use is unclear.

- Referencing needs addressing as often statements are made without reference.

Abstract:

- There is a lack of clarity about what the Laab Nuer model is, who it is for.

- I cannot identify the research problem that this research is looking to address

Intro & Method.

- I found the introduction lacks clarity and organisation. It would benefit from having headings and making sure it has a logical flow.

- You need to start with the research problem, and what you are trying to address.

- Whilst there is a Figure 1 which looks at the three research areas, without the research problem being clearly identified this is not as useful as it could be.

Method :

- There is reference to 'conferring with a restaurant owner' - is this part of the research protocol, what ethical approval processes were used. This seems colloquial and needs accurate academic referencing.

- The limitations need to be better explained, and what was done to overcome them.

Model:

- I found the model hard to follow, because I am not sure who it is for and how it answers your research questions.

- Is this for policymakers, is it for hospitality?

Discussion

- You write that: Thus, this model succeeds effectively in managing food safety in traditional Lanna cuisine. but that also "Therefore, the "Laab Nuer Model" needs to be researched". This is not clear.

**Do you want your identity to be public for this peer review?** For information about this choice, including consent withdrawal, please see our Privacy Policy

Reviewer #1: **Yes: ** Amirreza Rezaei

Reviewer #2: No

---

## [Author Response · Author response to Decision Letter 1]

4 Feb 2025

Dear Editors

We wish to revision version of the manuscript entitled “The Development of the “Laab Nuer Model” for food safety management in handling traditional Lanna cuisine at Thailand.” for consideration by the a PLOs One Journal .

We certify that we have implemented improvements and modifications to the article as recommended by the editor and the reviewers as a summary information below;

1. We revise abstract section to mention the problem and the main objective of this study.

2. We reorganize the flow of the introduction based on both reviewers.

3. We provide the information on the method section to clearly methodology of this study.

4. We change the part of The final model for food safety management for traditional Lanna cuisine to more describe the model creation and the implementation.

5. We add the limitation as a recommendation as the additional study, and update the final paragraph of discussion to clearly anticipate for the audience

Please address all correspondence concerning this manuscript to me at vivat.kea@mfu.ac.th

Thank you for your consideration of this manuscript.

Sincerely,

Vivat Keawdounglek.

---

## [Decision Letter · Decision Letter 1]

17 Apr 2025

Dear Dr. KEAWDOUNGLEK,

Thank you for submitting your manuscript to PLOS ONE. After careful consideration, we feel that it has merit but does not fully meet PLOS ONE’s publication criteria as it currently stands. Therefore, we invite you to submit a revised version of the manuscript that addresses the points raised during the review process.

**One or more of the reviewers has recommended that you cite specific previously published works. Members of the editorial team have determined that the works referenced are not directly related to the submitted manuscript. As such, please note that it is not necessary or expected to cite the works requested by the reviewer**

We look forward to receiving your revised manuscript.

Kind regards,

Mahmood Ahmed

Academic Editor

PLOS ONE

Reviewers' comments:

Reviewer's Responses to Questions

**Comments to the Author**

Reviewer #3: All comments have been addressed

Reviewer #4: (No Response)

2. Is the manuscript technically sound, and do the data support the conclusions?

Reviewer #3: Yes

Reviewer #4: Partly

3. Has the statistical analysis been performed appropriately and rigorously?

Reviewer #3: Yes

Reviewer #4: N/A

4. Have the authors made all data underlying the findings in their manuscript fully available?

Reviewer #3: Yes

Reviewer #4: No

5. Is the manuscript presented in an intelligible fashion and written in standard English?

Reviewer #3: Yes

Reviewer #4: No

Reviewer #3: This research effectively addresses a critical issue in food safety management, specifically in the context of traditional Lanna cuisine. The proposed "Laab Nuer Model" is a notable innovation, offering a comprehensive framework for addressing food safety across the entire supply chain—from raw material sourcing (upstream) to consumer education (downstream). Below is a detailed review with strengths and actionable suggestions for improvement.

Strengths of the Manuscript

1. Innovative Framework:

The "Laab Nuer Model" integrates three key stages—upstream (markets), midstream (restaurants), and downstream (consumers)—into a cohesive approach. This structured model ensures that food safety is addressed holistically, covering chemical contamination, hygiene practices, and consumer awareness.

2. Scientific Rigor:

The use of standardized protocols (e.g., HPLC for Paraquat analysis and MPN for biological contamination) demonstrates the technical robustness of the study.

The sample sizes were determined appropriately using G*Power, ensuring adequate statistical power, and reliability was assessed using Cronbach’s Alpha, which adds confidence to the findings.

Statistical tests such as One-Way ANOVA were performed rigorously to validate the significance of the model’s components.

3. Practical Applicability:

The model emphasizes actionable interventions, such as incentivizing farmers to follow good agricultural practices (GAP), conducting hygiene training for food handlers, and educating consumers. These measures are well-aligned with real-world implementation.

4. Data Transparency:

The authors have made the data underlying their findings fully available, in compliance with PLOS ONE’s data-sharing policy. This openness enhances the reproducibility and credibility of the research.

5. Focus on Regional Relevance:

The emphasis on Lanna cuisine reflects the cultural and economic importance of food safety in northern Thailand, with potential implications for the tourism industry. This localized focus is a strength, as it provides a practical solution tailored to specific cultural practices.

Suggestions for Improvement

1. Clarity and Language:

While the manuscript is intelligible, certain sections (e.g., Introduction and Discussion) contain awkward phrasing and minor grammatical errors. For instance:

Phrases like "food includes nutrients and minerals that are essential for the body's growth and repair" could be simplified to "food provides essential nutrients for growth and repair."

Consider revising sentences in the Discussion section for conciseness and flow.

2. Figures and Visual Representation:

Some figures (e.g., Paraquat concentration comparisons in different types of plants and E. coli contamination data) could be enhanced with clearer labels and legends to make the data easier to interpret.

Graphs should include units, axis titles, and descriptive captions that explain the significance of the data.

3. Expand on Generalizability:

While the study focuses on Lanna cuisine, a brief discussion of how the model could be adapted to other regions or food types would enhance its broader applicability. For example:

How might this model address similar challenges in street food markets or other traditional cuisines globally?

Could the upstream interventions, such as GAP certification, be adapted to other agricultural supply chains?

4. Statistical Reporting:

The statistical analysis could benefit from more detailed reporting. For instance, including confidence intervals, effect sizes, or post-hoc analysis results would provide a deeper understanding of the findings.

Adding a summary table of the key statistical results in the Results section might improve clarity.

5. Consumer Education and Awareness:

The downstream recommendations could be expanded to include modern approaches such as leveraging social media campaigns, mobile apps, or community workshops to educate consumers on food safety.

The manuscript could also address potential challenges in changing consumer behavior, particularly in rural or low-literacy populations.

6. Limitations and Future Research:

The limitations of the study are not explicitly addressed. For example:

Were there any challenges in obtaining reliable data or ensuring compliance during the implementation of interventions?

Could other potential contaminants (e.g., heavy metals, aflatoxins) also be relevant in the context of Lanna cuisine?

Future research directions should be outlined more clearly. For instance, exploring how the model performs when scaled up to larger regions or adapted for international use could be valuable.

Conclusion

The manuscript presents a robust and well-thought-out framework for improving food safety in traditional Lanna cuisine. The proposed "Laab Nuer Model" is technically sound and supported by appropriate data and statistical analysis. With its focus on practical interventions and local relevance, the study has the potential to significantly reduce foodborne illnesses in northern Thailand while also serving as a model for other regions and food types.

To further strengthen the manuscript, the authors should focus on improving clarity, enhancing visual presentation, and expanding the discussion of generalizability and limitations. These adjustments will not only improve the readability of the manuscript but also increase its impact and relevance to a broader audience.

This study is a valuable contribution to the field of food safety and is suitable for publication after minor revisions.

Reviewer #4: Review Comments to the Author

Thank you for the opportunity to review your manuscript titled “The Development of the Laab Nuer Model for food safety management in handling traditional Lanna cuisine at Thailand.” The manuscript presents an important and culturally relevant initiative in food safety management through a structured three-level approach (upstream–midstream–downstream) rooted in the context of Lanna cuisine in Chiang Rai, Thailand.

Your study offers valuable practical insights, especially with its use of real-world contamination tests (e.g., Paraquat and E. coli), and the community-based model you propose has significant potential for public health and local food system improvement.

However, the manuscript requires several important revisions to improve its clarity, scientific rigor, and compliance with PLOS ONE editorial standards:

1. Technical Soundness and Methodology

While your conceptual design is thoughtful, the sampling strategies, tool validation, and detailed methodology need further elaboration. For example, details about how participants (consumers, restaurants, market vendors) were selected, and how questionnaires were developed and validated (e.g., Cronbach’s alpha, factor analysis) are missing or underexplained. Additionally, claims regarding contamination reduction or consumer behavior impact should be supported with more robust statistical comparisons.

2. Statistical Analysis

The statistical analysis is appropriate in general terms (e.g., One-Way ANOVA for consumer survey data), but its application is not rigorous enough. Assumptions (normality, homogeneity), effect sizes, confidence intervals, and post-hoc tests are not addressed. Moreover, the E. coli results are promising but reported descriptively; they would benefit from a formal statistical test (e.g., paired t-test) to confirm significance.

3. Data Availability

Currently, it is unclear whether all underlying raw data (e.g., individual sample results, questionnaire responses) are publicly accessible or included in supplementary files. According to PLOS data policy, authors are required to make all relevant data fully available without restriction. Please clarify this in a formal Data Availability Statement and ensure all datasets are either uploaded or linked to a public repository.

4. Language and Presentation

The manuscript needs substantial English language revision. There are frequent grammatical errors, awkward phrasing, and terminology misuse (e.g., “consume” instead of “consumer level”). Figures and tables also require improved resolution and clearer captions. Professional language editing is highly recommended to ensure clarity and accessibility to an international audience.

5. Model Justification and Broader Context

While the “Laab Nuer Model” is a promising localized framework, its theoretical foundation and originality should be more clearly articulated. Why is this model unique, and how does it differ from existing food safety frameworks (e.g., HACCP)? Additionally, the discussion would be enriched by engagement with international literature to contextualize the model globally.

6. Application and Scalability

The conclusion mentions the potential application of this model in other contexts such as supermarkets or flea markets. However, no concrete strategies or considerations are provided. Discussing barriers, facilitators, and adaptation strategies would significantly enhance the generalizability of your findings.

Final Comments:

This is an important and timely contribution to food safety research in traditional food systems. I encourage the authors to revise the manuscript with close attention to scientific transparency, linguistic clarity, and statistical robustness. With these improvements, the paper will offer strong value for both public health researchers and food policy stakeholders.

**Do you want your identity to be public for this peer review?** For information about this choice, including consent withdrawal, please see our Privacy Policy

Reviewer #3: No

Reviewer #4: No

---

## [Author Response · Author response to Decision Letter 2]

19 May 2025

May 17, 2025

Dear Editors

We wish to revision version of the manuscript entitled “The Development of the “Laab Nuer Model” for food safety management in handling traditional Lanna cuisine at Thailand.” for consideration by the a PLOs One Journal .

We certify that we have implemented improvements and modifications to the article as recommended by the editor and the reviewers (as can be seen in Table A).

Please address all correspondence concerning this manuscript to me at vivat.kea@mfu.ac.th

Thank you for your consideration of this manuscript.

Sincerely,

Vivat Keawdounglek.

---

## [Decision Letter · Decision Letter 2]

26 Jun 2025

Dear Dr. KEAWDOUNGLEK,

Thank you for submitting your manuscript to PLOS ONE. After careful consideration, we feel that it has merit but does not fully meet PLOS ONE’s publication criteria as it currently stands. Therefore, we invite you to submit a revised version of the manuscript that addresses the points raised during the review process.

We look forward to receiving your revised manuscript.

Kind regards,

Karthikeyan Venkatachalam, Ph.D.

Academic Editor

PLOS ONE

Reviewers' comments:

Reviewer's Responses to Questions

**Comments to the Author**

Reviewer #3: All comments have been addressed

Reviewer #4: All comments have been addressed

Reviewer #5: (No Response)

2. Is the manuscript technically sound, and do the data support the conclusions?

Reviewer #3: Yes

Reviewer #4: Partly

Reviewer #5: Partly

3. Has the statistical analysis been performed appropriately and rigorously?

Reviewer #3: Yes

Reviewer #4: Yes

Reviewer #5: No

4. Have the authors made all data underlying the findings in their manuscript fully available?

Reviewer #3: Yes

Reviewer #4: Yes

Reviewer #5: Yes

5. Is the manuscript presented in an intelligible fashion and written in standard English?

Reviewer #3: Yes

Reviewer #4: Yes

Reviewer #5: Yes

Reviewer #3: The revised manuscript has adequately addressed all previous comments and concerns. The authors have provided a comprehensive and well-structured study on the development of the "Laab Nuer Model" for food safety management in traditional Lanna cuisine. The methodology is sound, the data supports the conclusions, and the statistical analysis has been performed rigorously. The manuscript is written in clear and standard English, and all underlying data have been made available as per PLOS ONE's data policy. The study contributes valuable insights into food safety management and aligns with global best practices. The manuscript is now acceptable for publication. Within my own research field, I can't offer any more opinions on this.

Reviewer #4: Reviewer Comments to the Author

Dear Authors,

I appreciate your effort in revising the manuscript. The revised version demonstrates improvement in structure and clarity, and addresses some of the concerns previously raised. However, several important issues remain either only partially resolved or insufficiently addressed, and further revisions are necessary before the manuscript can be considered for publication.

1. Language and Clarity

Although minor language edits were applied throughout the manuscript, significant issues with grammar, sentence construction, and scientific clarity remain (e.g., use of terms such as “consume level,” “may be received knowledge”). These affect the intelligibility of the work.

Recommendation: Please seek professional English editing service to ensure clarity and fluency appropriate for international publication.

2. Conceptual Framework of the Laab Nuer Model

You have now defined the Laab Nuer Model in terms of upstream, midstream, and downstream components, and included a conceptual diagram. However, it remains unclear why this specific dish (“Laab Nuer”) was chosen to model general food safety behavior. The cultural justification and theoretical rationale are still insufficiently developed.

Recommendation:

Elaborate on the theoretical foundation of the model and its potential for broader application.

Discuss how this model aligns or differs from internationally recognized frameworks such as HACCP or WHO guidelines.

Strengthen conceptual integration in both Methods and Discussion sections.

3. Methodological Transparency

Improvements have been made in explaining the use of purposive sampling, and some methodological details have been added (e.g., IOC, Cronbach's alpha). However:

The criteria for selecting restaurants using purposive sampling remain vague.

The actual questionnaire remains inaccessible, and response items are not described.

The data collection instruments and their validation still lack detailed documentation.

Recommendation:

Include clear selection criteria for participants and settings.

Provide the questionnaire as supplementary material.

Describe how validity and reliability were assessed.

4. Statistical Reporting

The application of one-way ANOVA is noted, but statistical assumptions (normality, homoscedasticity) are not evaluated. Also, effect sizes or confidence intervals are not reported, reducing the robustness of the interpretation.

Recommendation:

Explicitly test and report assumptions for ANOVA.

Include effect sizes (e.g., eta-squared) and interpret findings accordingly.

5. Discussion and International Context

The manuscript continues to emphasize local observations without adequately situating them within international food safety literature. Global guidelines (FAO/WHO, HACCP) are minimally referenced and not critically compared to the study’s findings.

Recommendation:

Integrate relevant international food safety studies.

Discuss how your findings align with or diverge from established global practices.

6. Ethics and Data Availability

While ethical approval is mentioned and verbal informed consent is reported, there is no justification for why written consent was not obtained. The PLOS ONE data sharing policy also appears insufficiently addressed.

Recommendation:

Justify the use of verbal rather than written consent.

Clearly state where and how the underlying data are available or justify any restrictions.

Summary Recommendation: Major Revisions

The manuscript contributes a culturally grounded framework to address food safety practices in traditional cuisine settings, which is commendable. However, to meet PLOS ONE publication standards, the manuscript requires substantial improvements in conceptual development, methodological detail, statistical rigor, and alignment with international standards.

Reviewer #5: Research Topic: The development of the “Laab Nuer Model” for food safety management

in handling traditional Lanna cuisine at Thailand

Reviewer comments:

In my opinion, the revised manuscript entitled 'The development of the Laab Nuer Model for food safety management in handling traditional Lanna cuisine at Thailand' is quite comprehensive. However, it is necessary to revise the text further before it can be published.

1. As I cannot see whether the authors have comprehensively addressed the previous reviewer’s comments, I have a question about this revised version: what is the rationale for determining this sample size?

2. Line 94 and 209: The term E. coli and EMB agar should be written out in full initially before using the abbreviated form.

3. Please carefully check the reference format in both reference lists and content following the journal format throughout the manuscript.

4. Equipment such as HPLC and column should include the manufacturer's brand, model, city, and country of origin.

5. All test kits should include the manufacturer's brand, city, and country of origin.

6. A statistical analysis section should be included to describe the experimental design, number of analytical replicates, and variance analysis methods, and the analytical tools employed.

7. Figures 3 and 4 should display error bars, and if statistical analysis indicates significant differences, these should be marked with appropriate symbols.

**Do you want your identity to be public for this peer review?** For information about this choice, including consent withdrawal, please see our Privacy Policy

Reviewer #3: No

Reviewer #4: No

Reviewer #5: No

---

## [Author Response · Author response to Decision Letter 3]

11 Aug 2025

Reviewer #4

(Green Hi-light)

4.1 Language and Clarity

Although minor language edits were applied throughout the manuscript, significant issues with grammar, sentence construction, and scientific clarity remain (e.g., use of terms such as “consume level,” “may be received knowledge”). These affect the intelligibility of the work.

Recommendation: Please seek professional English editing service to ensure clarity and fluency appropriate for international publication.

The revision

Thank you for your initial comments regarding the language and clarity of the manuscript. Based on your recommendation, we submitted the manuscript for professional proofreading by an experienced academic editor from the United Kingdom. This revised version has already been proofread by this professional editing service. In addition, the expert of English editing suggest that the name of article should be present as the entitle “The Development of the “Laab Nuer Model” for food safety management in handling traditional Lanna cuisine in Thailand ”

4.2 Conceptual Framework of the Laab Nuer Model

You have now defined the Laab Nuer Model in terms of upstream, midstream, and downstream components, and included a conceptual diagram. However, it remains unclear why this specific dish (“Laab Nuer”) was chosen to model general food safety behavior. The cultural justification and theoretical rationale are still insufficiently developed.

Recommendation: Elaborate on the theoretical foundation of the model and its potential for broader application. Discuss how this model aligns or differs from internationally recognized frameworks such as HACCP or WHO guidelines. Strengthen conceptual integration in both Methods and Discussion sections.

The revision:

From this valuable recommendation, I revise this information as list follows;

- Expanded the explanation of how Laab Nuer represents a culturally embedded, high-risk traditional dish with symbolic importance and widespread consumption in Northern Thailand. (Line 110-122)

- Described the model’s structure using the upstream–midstream–downstream framework to mirror the food safety risks and practices across the supply chain, in line with the farm-to-fork approaches. (Line 123-127)

- Clarified alignment with international standards, including HACCP principles and WHO food safety guidelines, while emphasizing how the model integrates local practices and informal food systems. (Line 129-135)

- Strengthened conceptual coherence across the Methods and Discussion sections by explicitly connecting the Laab Nuer Model with the European Commission's farm-to-fork strategy and the WHO's guidance on safe and culturally appropriate food systems. (Line 508-519, 520-525)

4.3 Methodological Transparency

Improvements have been made in explaining the use of purposive sampling, and some methodological details have been added (e.g., IOC, Cronbach's alpha). However: The criteria for selecting restaurants using purposive sampling remain vague. The actual questionnaire remains inaccessible, and response items are not described. The data collection instruments, and their validation still lack detailed documentation.

Recommendation: Include clear selection criteria for participants and settings. Provide the questionnaire as supplementary material. Describe how validity and reliability were assessed.

The revision

Following the reviewer’s recommendation, we have added the inclusion criteria for the downstream study in lines 224-228. A purposive sample size diagram for traditional Lanna restaurants has also been constructed and is now presented in Fig. 3 at the page of 7. In addition, the questionnaire used for the customer survey has been provided as Supplementary Material B (see line 260). Furthermore, we have explained the procedures for content validity and reliability assessment, including IOC and Cronbach’s alpha analysis, in lines 272-282.

4.4 Statistical Reporting

The application of one-way ANOVA is noted, but statistical assumptions (normality, homoscedasticity) are not evaluated. Also, effect sizes or confidence intervals are not reported, reducing the robustness of the interpretation.

Recommendation: Explicitly test and report assumptions for ANOVA. Include effect sizes (e.g., eta-squared) and interpret findings accordingly.

The revision

From the valuable recommendation, we add the calculation of normality by the Kolmogorov–Smirnov test (line 286-287), and we report the Levene’s test for homogeneity of variances as can be seen in the Supplementary Material D. The effect size of the ANOVA analysis has been added up in the methodology and the result. For the effect size interpretation, we select the effect size interpretation by the study of Richardson (2011), which indicated that an effect size greater than 0.14 is a large effect (Line 284-285, 367-368).

4.5 Discussion and International Context

The manuscript continues to emphasize local observations without adequately situating them within international food safety literature. Global guidelines (FAO/WHO, HACCP) are minimally referenced and not critically compared to the study’s findings.

Recommendation: Integrate relevant international food safety studies. Discuss how your findings align with or diverge from established global practices.

The revision

Thank you for the recommendation. We have changed the discussion section to make it clear how the Laab Nure Model corresponds with international food safety standards, such as the World Health Organization (WHO) guidelines, the Codex Alimentarius standards, the One Health concept, and the HACCP framework. While the Laab Nure Model does not emphasize formal documentation or regulatory compliance in the same way as HACCP, it instead focuses on enhancing local knowledge and awareness of food safety within the context of traditional cuisine. These revisions can be found in lines 508-532 of the revised manuscript.

4.6 Ethics and Data Availability

While ethical approval is mentioned and verbal informed consent is reported, there is no justification for why written consent was not obtained. The PLOS ONE data sharing policy also appears insufficiently addressed.

Recommendation: Justify the use of verbal rather than written consent. Clearly state where and how the underlying data are available or justify any restrictions.

The revision:

We apologize for this mistake on the ethics and data availability. In fact, we use the written informed consent form for all participants of this study that was approved by the Chiang Rai Provincial Health Office (experimental protocol no. CRPPHO 131/2564). Moreover, we mention that all participants have written the informed consent form for this study, as can be seen in lines 168-169.

Reviewer #5

(Yellow Hi-light)

5.1 As I cannot see whether the authors have comprehensively addressed the previous reviewer’s comments, I have a question about this revised version: what is the rationale for determining this sample size?

The revision:

Thak you for your valuable recommendation. From this address, we can explain the reason of sample size collection as lists follows;

1) The study area is mentioned at Chiang Rai, Thailand, because this area is the origin of traditional Lanna Cuisine related to the study of Onanong Thongmee et al. (Line 80-85)

2) The target group of upstream is the Fresh market in Chiang Rai, which is the biggest fresh market in Chiang Rai, and it is a big source of raw material distribution. (Line 176-178)

3) The target group of midstram: we have added the inclusion criteria for the midstream study in lines 224-228. A purposive sample size diagram for traditional Lanna restaurants has also been constructed and is now presented in Fig. 3. (Page 7)

4) The target group of downstream, we collect from the 305 customers who have ever experienced traditional Lanna Cuisine. Moreover, G*Power is used for the sample calculation for the downstream operation. (Page 8)

5.2 Line 94 and 209: The term E. coli and EMB agar should be written out in full initially before using the abbreviated form.

The revision:

From this concern, we revise for the full inially as the before using the abbreviated form whereas the full name of Escherichia coli is illustrated at line 93, and the full name of Eosin Methylene Blue (EMB) agar is displayed at the line 230.

5.3 Please carefully check the reference format in both reference lists and content following the journal format throughout the manuscript.

The revision:

We check the reference format for both of intext citation and the list of reference according to the PLOs One Guildeline manual for the intext citation and reference writing.

5.4 Equipment such as HPLC and column should include the manufacturer's brand, model, city, and country of origin.

The revision :

From this recommendation, we add the comericial name of HPLC column as the SunFire TM C-18 column made in Ireland ( line 187-188, 190-191) , and the company of Himedia Co.Ltd, for the EMB agar (Line 230).

5.5 All test kits should include the manufacturer's brand, city, and country of origin.

The revision:

From this recommendation, we add the commercial name of the chemical test-kit called as the GPO, and this test-kit made in Thailand. This information is provided as the line 204-210.

5.6 A statistical analysis section should be included to describe the experimental design, number of analytical replicates, and variance analysis methods, and the analytical tools employed.

The revision:

Thank you for your important recommendation; we add the “cross-sectional design” for the study design as a line of 155. For the laboratory analysis, including the pesticide, chemical, and biological contamination, we have a triplicate for this analysis, as can be seen in lines 210-211 and 239 . In addition, we revise the data interpretation in the downstream using the IBM SPSS for analysis of the average, p-Value of the ANOVA test, the Levene’s test for homogeneity of variances ,and the effect size of factors from the Eta Square calculation, as can be seen in lines 283-287 and the supplementary material D.

5.7 Figures 4 (3) and 5(4) should display error bars, and if statistical analysis indicates significant differences, these should be marked with appropriate symbols.

The revision:

For this valuable recommendation, we create the error bar for fig. 4 as the 9’s page . However, in fig. 5 is the comparison of E.coli contamination from the pre-training and post-training at the midstream. Therefore, there is no evidence to support the error bars from SPSS associated with the chemical, and biological contamination. In addition, The p-Value at the downstream was changed to the appropriate symbols; especially the ANOVA score was then changed from p-Value = 0.00 to p-Value < 0.05 as can be seen in page 11.

---

## [Decision Letter · Decision Letter 3]

25 Aug 2025

The Development of the “Laab Nuer Model” for food safety management in handling traditional Lanna cuisine in Thailand

PONE-D-24-37780R3

Dear Dr. KEAWDOUNGLEK,

We’re pleased to inform you that your manuscript has been judged scientifically suitable for publication and will be formally accepted for publication once it meets all outstanding technical requirements.

Kind regards,

Karthikeyan Venkatachalam, Ph.D.

Academic Editor

PLOS ONE

Additional Editor Comments (optional):

Reviewers' comments:

Reviewer's Responses to Questions

**Comments to the Author**

Reviewer #5: All comments have been addressed

2. Is the manuscript technically sound, and do the data support the conclusions?

Reviewer #5: Yes

3. Has the statistical analysis been performed appropriately and rigorously?

Reviewer #5: Yes

4. Have the authors made all data underlying the findings in their manuscript fully available?

Reviewer #5: Yes

5. Is the manuscript presented in an intelligible fashion and written in standard English?

Reviewer #5: Yes

Reviewer #5: 5.1 As I cannot see whether the authors have comprehensively addressed the previous reviewer’s comments, I have a question about this revised version: what is the rationale for determining this sample size?

Recommendation: the comment has been addressed.

5.2 Line 94 and 209: The term E. coli and EMB agar should be written out in full initially before using the abbreviated form.

Recommendation: the comment has been addressed.

5.3 Please carefully check the reference format in both reference lists and content following the journal format throughout the manuscript.

Recommendation: the comment has been addressed.

5.4 Equipment such as HPLC and column should include the manufacturer's brand, model, city, and country of origin.

Recommendation: the comment has been addressed.

5.5 All test kits should include the manufacturer's brand, city, and country of origin.

Recommendation: the comment has been addressed.

5.6 A statistical analysis section should be included to describe the experimental design, number of analytical replicates, and variance analysis methods, and the analytical tools employed.

Recommendation: the comment has been addressed.

5.7 Figures 4 (3) and 5(4) should display error bars, and if statistical analysis indicates significant differences, these should be marked with appropriate symbols.

The revision:

Recommendation: the comment has been addressed.

**Do you want your identity to be public for this peer review?** For information about this choice, including consent withdrawal, please see our Privacy Policy

Reviewer #5: No

---

## [Editor Report · Acceptance letter]

PONE-D-24-37780R3

PLOS ONE

Dear Dr. Keawdounglek,

I'm pleased to inform you that your manuscript has been deemed suitable for publication in PLOS ONE. Congratulations! Your manuscript is now being handed over to our production team.

Kind regards,

on behalf of

Dr. Karthikeyan Venkatachalam

Academic Editor

PLOS ONE